# Synergistic Use of Thermostable Laccase and Xylanase in Optimizing the Pre-Bleaching of Kraft Pulp

**Kartik Patel [1,2], Nilam Vaghamshi [2], Kamlesh Shah [3], Srinivas Murty Duggirala [4], Anjana Ghelani [5], Pravin Dudhagara [2] and Douglas J. H. Shyu [6,*]**

1   Biotech Research and Development Lab, Witmans Industries Private Limited, Bhimpore, Daman 396210, India; kvpatel@vnsgu.ac.in
2   Department of Biosciences (UGC-SAP-DRS-II & DST-FIST-I), Veer Narmard South Gujarat University, Surat 395007, India; ndvaghmashi@vnsgu.ac.in (N.V.); pravindudhagara@vnsgu.ac.in (P.D.)
3   Department of Microbiology, R. G. Shah Science College Vasana, Ahmedabad 380007, India; kamleshkumar.shah01@gujgov.edu.in
4   Biogas Research Centre, P. G. Department of Microbiology, M. D. Gramsewa Mahavidyalaya, Gujarat Vidyapith Sadra, Gandhinagar 382320, India; dsrinivasmurty@gujaratvidyapith.org
5   Shree Ramkrishna Institutes of Computer Education & Applied Sciences, Sarvajanik University, Surat 395001, India; anjana.ghelani@srki.ac.in
6   Department of Biological Science and Technology, National Pingtung University of Science and Technology, Neipu, Pingtung 912301, Taiwan
*   Correspondence: dshyu@mail.npust.edu.tw

**Abstract:** The continuous requirement for pre-bleaching processes on kraft pulp, employing a range of compatible enzymes, aims to mitigate the pollution caused by chemical bleaching agents. In the present study, the laccase-producing bacterium *Bacillus licheniformis* BK-1 was isolated from the Bakreshwar hot spring in India and tested for laccase production using different lignocellulosic substrates. The isolate was found to produce maximum laccase (8.25 IU/mL) in the presence of rice bran as a substrate, followed by 5.14 IU/mL using sawdust over a 48 h period. Laccase production doubled when medium parameters were optimized using a central composite design. The bleaching of rice straw pulp was accomplished using a laccase, xylanase (previously extracted from the same bacteria), and laccase–xylanase mixture. The mix-wood kraft pulp treated with the enzyme mixture at pH 7.0 and 50 °C temperature for up to 180 min reduced the chlorine amount by 50% compared to the control. The results also revealed that the enzyme mixture improved the pulp's optical (brightness 10.39%) and physical (tear index 39.77%, burst index 22.82%, and tensile strength 14.28%) properties with 50% chlorine dose. These exceptional properties underscore the enzyme mixture's suitability for pulp pre-bleaching in paper manufacturing, offering a safer and more environmentally friendly process.

**Keywords:** laccase; xylanase; *Bacillus licheniformis*; hot spring; biobleaching; pre-bleaching

## 1. Introduction

The pulp and paper industry is one of the largest in the world, and India has experienced impressive growth in recent years as a country with a rapidly expanding economy. Simultaneously, the pulp and paper industries contribute to environmental pollution, which has become one of the world's major issues [1]. In order to maintain a pollution-free and clean environment, government policies are placing immense pressure on the paper industries [2]. Bleaching requires a large amount of chemicals that require optimization to reach high efficiency. Regardless of origin, pulp bleaching has used large amounts of chlorinated compounds to remove lignin from pulp, releasing highly toxic and bioaccumulating chlorinated organic substances in the effluent [3]. In response to this pressure, the pulp and paper industries are attempting to change and reduce chlorine-based chemicals and switch

to more environmentally friendly methods for reducing the use and emission of harmful chemicals without compromising paper quality [3].

Utilizing microbial enzymes for the biobleaching of pulp is often regarded as the most pertinent and environmentally conscious method. Hemicellulolytic enzymes, such as xylanase (EC 3.2.1.8), pectinase (EC 3.2.1.15), and mannanase (EC 3.2.1.78), play a crucial role in the depolymerization and elimination of hemicellulolytic constituents. This enzymatic activity enables the removal of lignin from the pulp with less chemical intervention [4,5]. In contrast, laccase enzymes (EC 1.10.3.2), classified as ligninolytic enzymes, exhibit a direct mode of action on lignin by targeting its phenolic subunits, resulting in its degradation [6]. Enzymatic action positively influences the kappa number, hexenuronic acid, chromophore compounds release, pulp crystallinity, morphology, and various other physical, chemical, and mechanical properties of pulp [7]. There has been a recent surge in interest surrounding lignin-removal enzymes, primarily driven by their significant potential and efficacy in bio-bleaching agro-based pulp at an industrial level [8,9]. However, fungal-derived enzymes such as xylanase, pectinase, and laccase have been predominantly used in previous pulp-bleaching application research. Nevertheless, a limited number of bacterial strains have been reported to possess hemicellulolytic and lignolytic capabilities [10,11]. Moreover, bacterial-origin enzymes are considered more valuable and advantageous than fungal sources due to their bio-prospecting capabilities for industrial applications and ease of modification [12]. *Bacillus* strains with tremendous thermal and alkaline stability have emerged as the desirable choice for the pulp and paper industry among the various bacteria that produce lignolytic and hemicellulolytic enzymes [1]. Additionally, it is worth noting that *Bacillus* strains that grow on inexpensive agro-waste substrates and exhibit enhanced production of extracellular enzymes offer additional advantages [13].

As a result of the progress made in enzyme technology, researchers are exploring the modification of the biobleaching process through the introduction of a mixture or consortium of enzymes during the pulp pre-bleaching stage. The enzymatic green approach was initially initiated with a single hydrolytic enzyme. However, revealing the lignolytic efficiency of laccase and compatibility with hydrolytic enzymes, the enzyme mixture concept has emerged, changing the biobleaching process's whole scenario [6]. The combination of hydrolytic enzymes facilitates the accessibility of residual lignin, eliminates hexenuronic acids, reduces the usage of mediators, and speeds up the procedure [1,4].

Based on the earlier evidence, a mixture of enzymes would be preferable to a single enzyme to effectively bleach cellulosic pulp. Previously, pulp bleaching was carried out by applying a mixture of xylanase–pectinase [1], laccase–mannanase [6], laccase–xylanase [14], xylanase–pectinase [5,15], and laccase–xylanase–mannanase [16]. Hence, the primary objective of this study is to assess the efficacy of laccase and xylanase enzymes during the pre-bleaching phase with the intention of reducing the number of chemicals required in the succeeding step. Additionally, we covered the production and optimization of enzymes from an affordable agro-waste substrate, the industry's most significant economic aspect.

## 2. Results and Discussion

### 2.1. Isolation and Identification of Laccase Producer

A total of eight morphologically distinct bacterial strains were isolated on nutrient agar from the Bakreshwar hot spring. Among the eight isolates, one isolate was designated as BK-1. It showed reddish-brown color colonies on the guaiacol agar plate, which indicates a positive producer and is also confirmed by streaking the same isolate on nutrient agar (Figure 1). Isolate BK-1 was identified as *Bacillus licheniformis* using 16S rRNA nucleotide sequencing. After identification, the nucleotide sequence was deposited in NCBI under accession number MW673663. Most Indian hot springs are dominated by different *Bacillus* species, including laccase-producing thermotolerant *B. licheniformis* [17,18].

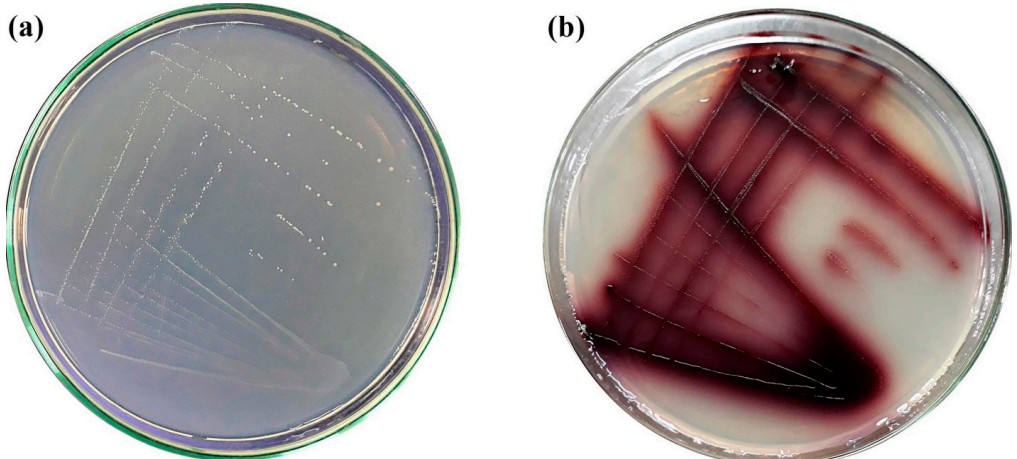

**Figure 1.** Screening of isolate BK-1 for laccase production. (**a**) Nutrient agar plate (control) and (**b**) guaiacol agar plate.

### 2.2. Production of Laccase on a Lignocellulosic Substrate via B. licheniformis BK-1

Agricultural wastes are numerous, and they are an extremely nutritious and rich source of carbohydrates [19]. Many bacterial species can utilize these diverse and inexpensive carbon wastes by producing ligninolytic enzymes and metabolizing the complicated residues of these waste substrates. In the current study, *B. licheniformis* BK-1 was able to produce laccase in the presence of a variety of agricultural waste materials (rice bran, wheat bran, barley bran, oat bran, sawdust, and citrus peel), with rice bran having the highest level of production (8.25 IU/mL), followed by sawdust (5.14 IU/mL), wheat bran (4.15 IU/mL), barley bran (3.84 IU/mL), oat bran (2.84 IU/mL), and citrus peel (1.25 IU/mL), respectively (Figure 2). The time–course experiment suggested that laccase production in the culture with rice bran increased gradually and reached its maximum point of 8.25 IU/mL at the 48 h time point. Laccase was reported to gradually diminish after 48 h due to proteolytic activity and autolysis of laccase. These findings are in accordance with the most recent bacterial laccase study from *Bacillus aquimaris* AKRC02, which found a laccase production of 4.58 IU/mL in the presence of rice bran in 120 h [20]. Unuofin [19] reported wheat bran as a powerful substrate for laccase (28.9 IU/mL) synthesis in *Bordetella* sp. JWO16 uses several agro-industrial wastes. Laccase production using low-cost substrates such as rice bran, wheat bran, and sugarcane molasses has also been described [21–23]. Secondly, considerable laccase synthesis utilizing sawdust as a substrate was reported, implying a chance to use sawdust created from the pulp and paper sectors, thus lowering sawdust waste and boosting the country's circular economy.

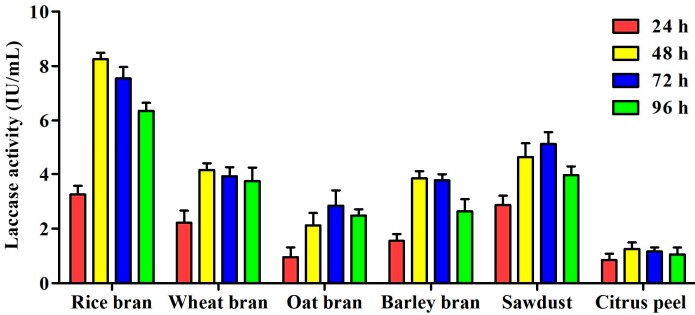

**Figure 2.** Time course of laccase production of *B. licheniformis* BK-1 with various lignocellulosic residues.

### 2.3. Optimization of Laccase Production Using Statistical Methods

#### 2.3.1. Screening of Important Components Using the Plackett–Burman (PB) Design

The interaction effect of 11 factors on the laccase production by *B. licheniformis* BK-1 was studied using a PB design approach. Different combinations (12 experimental runs)

showed a wide variation in the amount of laccase production measured in each run, ranging from 8.32 to 15.08 IU/mL in the medium used for enzyme production (Table 1). The influence (positive and negative effects) of different factors on enzyme production has been illustrated in the Pareto chart and half-normal plot (Figure 3). Of all these factors, rice bran, yeast extract, $CuSO_4 \cdot 5H_2O$, and temperature were the most influencing factors for the laccase.

**Table 1.** Design matrix and experimental results of the Plackett–Burman (PB) design for laccase production (minimum and maximum ranges of various factors are mentioned in each run).

| Run | A: Glucose (g/L) | B: Rice Bran (g/L) | C: Yeast Extract (g/L) | D: $K_2HPO_4$ (g/L) | E: $MgSO_4 \cdot 7H_2O$ (g/L) | F: NaCl (g/L) | G: $CuSO_4 \cdot 5H_2O$ (g/L) | H: $CaCl_2$ (g/L) | I: Temperature (°C) | J: Ph | K: Inoculum Size (%) | Laccase Activity (IU/mL) Actual | Predicted |
|---|---|---|---|---|---|---|---|---|---|---|---|---|---|
| 1 | 0.5 | 5.0 | 15.0 | 0.5 | 0.1 | 1.0 | 0.005 | 0.05 | 30 | 5.0 | 0.5 | 8.32 | 8.47 |
| 2 | 2.0 | 5.0 | 25.0 | 2.0 | 0.4 | 1.0 | 0.005 | 0.05 | 60 | 5.0 | 2.0 | 11.85 | 12.16 |
| 3 | 0.5 | 30.0 | 25.0 | 2.0 | 0.1 | 1.0 | 0.005 | 0.2 | 30 | 10.0 | 2.0 | 12.72 | 12.85 |
| 4 | 2.0 | 5.0 | 25.0 | 2.0 | 0.1 | 4.0 | 0.015 | 0.2 | 30 | 5.0 | 0.5 | 11.79 | 11.67 |
| 5 | 0.5 | 30.0 | 15.0 | 2.0 | 0.4 | 1.0 | 0.015 | 0.2 | 60 | 5.0 | 0.5 | 14.43 | 13.9 |
| 6 | 2.0 | 5.0 | 15.0 | 0.5 | 0.4 | 1.0 | 0.015 | 0.2 | 30 | 10.0 | 2.0 | 8.96 | 9.72 |
| 7 | 2.0 | 30.0 | 15.0 | 2.0 | 0.4 | 4.0 | 0.005 | 0.05 | 30 | 10.0 | 0.5 | 11.39 | 10.9 |
| 8 | 0.5 | 5.0 | 25.0 | 0.5 | 0.4 | 4.0 | 0.005 | 0.2 | 60 | 10.0 | 0.5 | 12.82 | 12.16 |
| 9 | 2.0 | 30.0 | 25.0 | 0.5 | 0.1 | 1.0 | 0.015 | 0.05 | 60 | 10.0 | 0.5 | 15.08 | 15.85 |
| 10 | 2.0 | 30.0 | 15.0 | 0.5 | 0.1 | 4.0 | 0.005 | 0.2 | 60 | 5.0 | 2.0 | 12.1 | 12.65 |
| 11 | 0.5 | 30.0 | 25.0 | 0.5 | 0.4 | 4.0 | 0.015 | 0.05 | 30 | 5.0 | 2.0 | 14.54 | 14.1 |
| 12 | 0.5 | 5.0 | 15.0 | 2.0 | 0.1 | 4.0 | 0.015 | 0.05 | 60 | 10.0 | 2.0 | 11.91 | 11.47 |

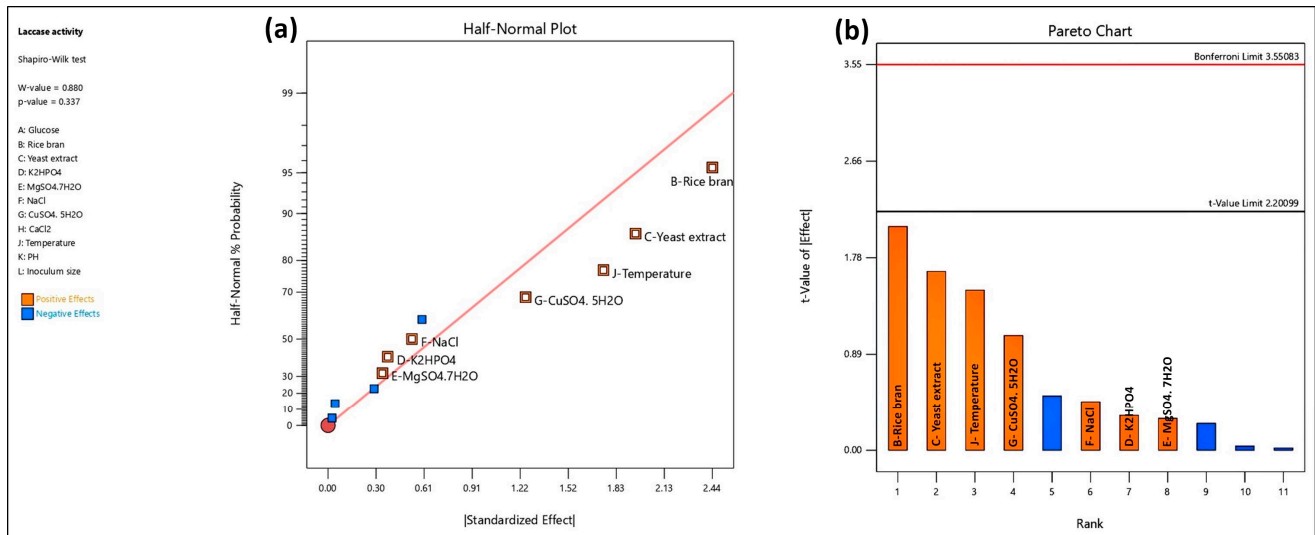

**Figure 3.** Construction of (**a**) a half-normal plot and (**b**) a Pareto chart during the screening of a medium component of laccase production by *B. licheniformis* BK-1 in the PB design. The orange color indicates positive effects, and the blue color suggests the negative effects of each parameter.

2.3.2. Optimization of Laccase Production via the Central Composite (CC) Design of Response Surface Methodology

In this response surface methodology (RSM) study, a CC design was used to determine the optimal levels of the four independent variables (rice bran, yeast extract, $CuSO_4 \cdot 5H_2O$, and temperature) to determine laccase production. The complete thirty experimental

matrixes, including the actual and predicted values of responses, are presented in Table 2. *B. licheniformis* BK-1 exhibited significant differences in laccase production, indicating that the process variables at all tested levels impact enzyme production. The medium containing 35 g/L of rice bran, 25 g/L of yeast extract 0.015 g/L $CuSO_4 \cdot 5H_2O$, and a temperature of 55 °C produced the maximum laccase (21.35 IU/mL), while the medium containing 27.5 g/L of rice bran, 10 g/L of yeast extract 0.02 g/L $CuSO_4 \cdot 5H_2O$, and with a temperature of 45 °C produced the least (3.98 IU/mL). A media formulation was optimized for maximal laccase production (run No. 20) in order to validate the media composition and its subsequent application in enzyme characterization and the pre-bleaching of kraft pulp. We utilized the replication of run no. 20 in order to quantify the equivalent activity of laccase (21.3 IU/mL).

**Table 2.** Design matrix and experimental results of central composite (CC) design.

| Run | A: Rice Bran | B: Yeast Extract | C: $CuSO_4 \cdot 5H_2O$ | D: Temperature | Laccase Activity (IU/mL) Actual | Laccase Activity (IU/mL) Predicted |
|---|---|---|---|---|---|---|
| 1 | 0 | 0 | −2 | 0 | 16.54 | 16.86 |
| 2 | 0 | 0 | 0 | 0 | 20.35 | 20.41 |
| 3 | 1 | −1 | −1 | 1 | 16.72 | 16.67 |
| 4 | 1 | −1 | −1 | −1 | 19.46 | 19.41 |
| 5 | 0 | 0 | 0 | 2 | 18.56 | 18.43 |
| 6 | −1 | −1 | 1 | 1 | 6.6 | 6.81 |
| 7 | 0 | 0 | 0 | 0 | 20.68 | 20.41 |
| 8 | 0 | −2 | 0 | 0 | 3.98 | 3.86 |
| 9 | 1 | 1 | 1 | −1 | 9.68 | 9.63 |
| 10 | −1 | 1 | 1 | −1 | 14.22 | 14.32 |
| 11 | −2 | 0 | 0 | 0 | 10.25 | 10.39 |
| 12 | −1 | 1 | −1 | −1 | 13.84 | 13.48 |
| 13 | −1 | −1 | −1 | 1 | 4.6 | 4.48 |
| 14 | 1 | 1 | −1 | 1 | 21.11 | 21.07 |
| 15 | 0 | 0 | 2 | 0 | 13.3 | 13.09 |
| 16 | 0 | 2 | 0 | 0 | 14.36 | 14.59 |
| 17 | 0 | 0 | 0 | 0 | 20.42 | 20.41 |
| 18 | 1 | 1 | −1 | −1 | 15.88 | 15.72 |
| 19 | −1 | −1 | 1 | −1 | 8.12 | 7.99 |
| 20 | −1 | 1 | 1 | 1 | 21.35 | 21.23 |
| 21 | 1 | −1 | 1 | 1 | 11.86 | 12.06 |
| 22 | 1 | 1 | 1 | 1 | 19.45 | 19.5 |
| 23 | 1 | −1 | 1 | −1 | 10.23 | 10.27 |
| 24 | 0 | 0 | 0 | 0 | 20.3 | 20.41 |
| 25 | 2 | 0 | 0 | 0 | 17.91 | 17.88 |
| 26 | −1 | 1 | −1 | 1 | 15.84 | 15.86 |
| 27 | 0 | 0 | 0 | 0 | 20.57 | 20.41 |
| 28 | 0 | 0 | 0 | −2 | 14.02 | 14.26 |
| 29 | −1 | −1 | −1 | −1 | 10.18 | 10.19 |
| 30 | 0 | 0 | 0 | 0 | 20.14 | 20.41 |

Note: Levels −2, −1, 0, 1, and 2 are the coded values of each variable. The actual value level of these coded values is given in Table S1 in the Supplementary File.

According to the RSM simulation, the quadratic model was the best approach for explaining the relationship between responses and factors. The empirical relationship between the independent factors and response (laccase production) was demonstrated using a quadratic equation, as shown in the following equation:

$$Y = +20.41 + 1.87\,A + 2.68\,B - 0.9417\,C + 1.04\,D - 1.74\,AB - 1.74\,AC + 0.7413\,AD + 0.7613\,BC + 2.02\,BD + 1.13\,CD - 1.57\,A^2 - 2.8\,B^2 - 1.36\,C^2 - 1.02\,D^2$$

where Y is the predicted response (laccase production), and A, B, C, and D are the coded values of the test factors: rice bran (g/L), yeast extract (g/L), $CuSO_4 \cdot 5H_2O$ (g/L), and temperature (°C), respectively.

A summary of the ANOVA in response to the surface quadratic model for the production of Laccase by *B. licheniformis* BK-1 is presented in Table 3. The model-generated F-value and *p*-value were calculated at 1041.2 and <0.0001, respectively, indicating that the model was significant. The lack of fit (LOF) for this process in the present model is insignificant (0.3034), indicating that the model had acceptable terms and adequate accuracy in data prediction. The model's accuracy and reliability were determined by evaluating predicted $R^2$ and an adequate precision [24]. The "Pred $R^2$" of 0.995 is consistent with the "Adj $R^2$" of 0.998.

**Table 3.** ANOVA for the CC design of the selected production parameters used in optimizing laccase production.

| Source | Sum of Squares | Df | Mean Square | F-Value | $p$ > (F) |
|---|---|---|---|---|---|
| Model | 780.7 | 14 | 55.76 | 1041.2 | <0.0001 *** |
| A—Rice bran | 84.23 | 1 | 84.23 | 1572.6 | <0.0001 *** |
| B—Yeast extract | 172.59 | 1 | 172.59 | 3222.54 | <0.0001 *** |
| C—$CuSO_4 \cdot 5H_2O$ | 21.28 | 1 | 21.28 | 397.36 | <0.0001 *** |
| D—Temperature | 26.04 | 1 | 26.04 | 486.24 | <0.0001 *** |
| AB | 48.65 | 1 | 48.65 | 908.38 | <0.0001 *** |
| AC | 48.23 | 1 | 48.23 | 900.58 | <0.0001 *** |
| AD | 8.79 | 1 | 8.79 | 164.14 | <0.0001 *** |
| BC | 9.27 | 1 | 9.27 | 173.12 | <0.0001 *** |
| BD | 65.37 | 1 | 65.37 | 1220.5 | <0.0001 *** |
| CD | 20.48 | 1 | 20.48 | 382.31 | <0.0001 *** |
| $A^2$ | 67.52 | 1 | 67.52 | 1260.68 | <0.0001 *** |
| $B^2$ | 214.5 | 1 | 214.5 | 4004.95 | <0.0001 *** |
| $C^2$ | 50.65 | 1 | 50.65 | 945.79 | <0.0001 *** |
| $D^2$ | 28.34 | 1 | 28.34 | 529.13 | <0.0001 *** |
| Residual | 0.8034 | 15 | 0.0536 | | |
| Lack of Fit | 0.6162 | 10 | 0.0616 | 1.65 | 0.3034 ns |
| Pure Error | 0.1872 | 5 | 0.0374 | | |
| Cor Total | 781.51 | 29 | | | |

Significant codes: $p$ < 0.001 ***; Not significant = ns; $R^2$ = 0.999; Adjusted $R^2$ = 0.998; Predicted $R^2$ = 0.995.

Figure 4a–f represent the profile of quadratic response surface plots for significant variable optimization (rice bran, yeast extract, $CuSO_4 \cdot 5H_2O$, and temperature). The 3D response surface plot shapes describe the importance of the interaction between corresponding parameters [25]. Each figure depicts the effect of two factors while holding the other factor constant. According to the statistics, laccase production initially increased significantly up to a certain level, and then it was gradually reduced with a higher value of significant components. The increase or decrease in enzyme production is also affected by the value of the carbon and nitrogen sources [22]. It is possible that the initial lower concentrations of these variables affect microbial growth, influencing the formation of the product. Because copper is a laccase inducer and part of this enzyme's active site, copper sulfate has a positive effect on laccase production. Copper sulfate concentrations above the recommended level (0.015 g/L) significantly inhibited laccase production. This is due to the inhibiting effect of copper at higher concentrations [26]. Furthermore, temperature is regarded as an important factor that significantly impacts enzyme production. According to the current findings, the maximum laccase production of *B. licheniformis* BK-1 was observed at 55 °C. In the present investigation, low temperature may reduce enzyme productivity because it lowers microbe efficiency, whereas high temperature may cause the thermal denaturation of the enzyme protein tertiary structure [27]. The experiment showed that after statistical optimization, laccase production was increased 2.58-fold, reaching 21.35 IU/mL from 8.25 IU/mL (unoptimized medium).

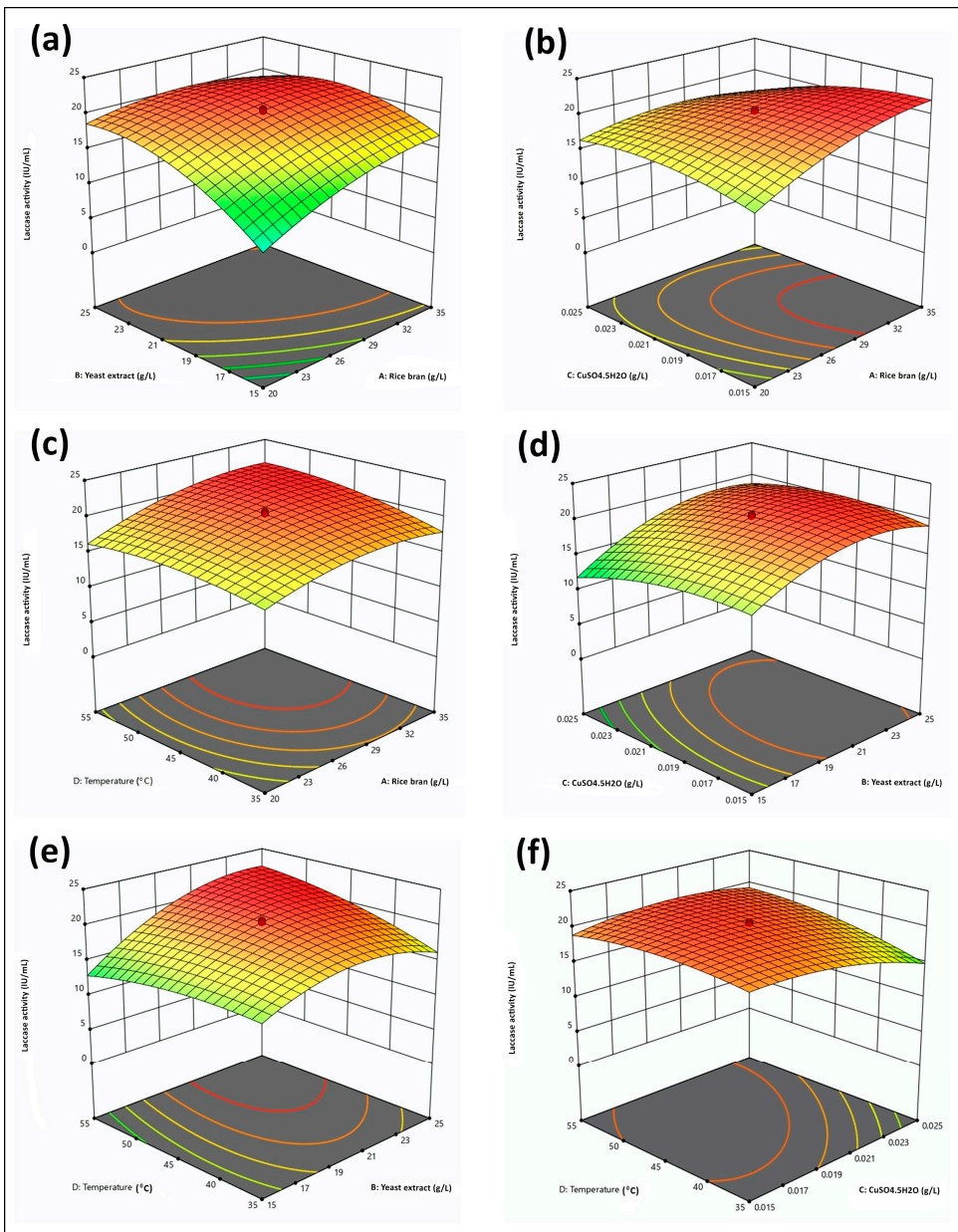

**Figure 4.** The 3D contour plot generated using CC design represents the effects of various media components and culture conditions for laccase production using BK: (**a**) rice bran and yeast extract; (**b**) rice bran and CuSO$_4$·5H$_2$O; (**c**) rice bran and temperature; (**d**) yeast extract and CuSO$_4$·5H$_2$O; (**e**) yeast extract and temperature; and (**f**) CuSO$_4$·5H$_2$O and temperature.

*2.4. Enzyme Characterization*

An enzyme must be active and stable under industrial process conditions in order to be used on a large scale [13]. Because the pulp is processed at elevated temperatures and pH levels in a paper mill [28], the pH and temperature profiles of the enzymes have been investigated. Based on the findings, the laccase of *B. licheniformis* BK-1 was identified as a potential choice for pulp bio-bleaching.

2.4.1. Effect of pH on Laccase Activity and Stability

The impact of pH on laccase activity and stability is shown in Figure 5. In the current study, laccase activity towards the substrate guaiacol peaked at pH 7.0, followed by pH 6.5, with a 96.74% relative activity at pH 7.0 (Figure 5a). Mehandia et al. [29] found that *Alcaligenes faecalis* laccase performs best at a pH of 8.0. The results were consistent with

studies by Kuddus et al. [30] and Lu et al. [31], which found that the pH range for bacterial laccase was best between pH 6.0 and 8.0. The laccase pH stability was investigated using guaiacol, and it was discovered that the laccase was most stable at pH 6.5, followed by pH 7.0. After 3 h of incubation, the enzyme retained 80.28% of its initial activity at pH 6.5 and 74.88% at pH 7.0 (Figure 5b). Laccase neutral activity and stability are comparable to those of previously studied bacterial laccases [32]. Biotechnological applications require laccase stability above pH 7.0, but only a few bacterial and fungal laccases are active above pH 7.0 [33]. The stability of enzymes in a wide range of pH may be due to the presence of more ionizable amino acid residues and hydrophobic interactions.

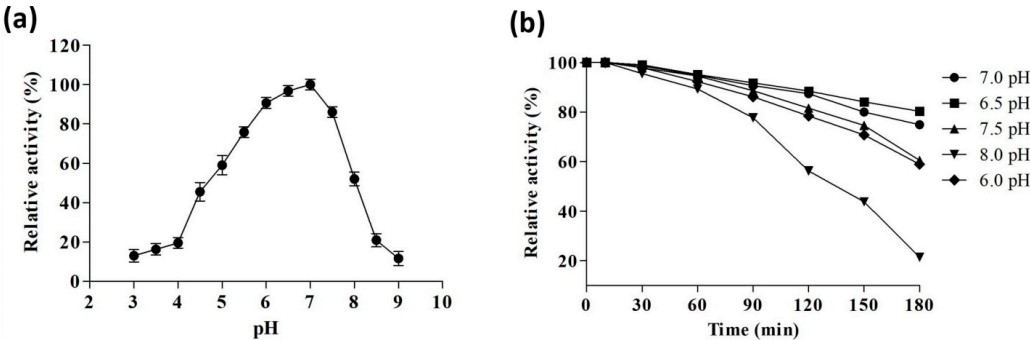

**Figure 5.** Effect of pH on the activity (**a**) and stability (**b**) of the laccase from BK-1 at 45 °C. (**a**) Laccase activity was measured at different pHs (pH 3.0–9.0) with guaiacol. (**b**) Residual activity was measured after the pre-incubation of the enzyme solution at pH 6.

### 2.4.2. Effect of Temperature on Laccase Activity and Stability

The impact of temperature on laccase activity and stability is shown in Figure 6. The laccase activity in the current study peaked at 55 °C towards the substrate guaiacol (Figure 6a). The thermal stability of laccase was examined using guaiacol, and it was found that the laccase was most stable at 45 °C, followed by 50 °C. The enzyme retained 94.12% of its initial activity at 45 °C and 91.25% at 50 °C after 3 h of incubation (Figure 6b). These findings are in agreement with some recently discovered bacterial laccases that demonstrated remarkable thermo-stability above 55 °C, such as the laccase from *A. faecalis* 9 [29], *Kocuria* sp. PBS-1 [10], *B. tequilensis* SN4 [34], and *B. licheniformis* LS04 [30], which are more thermostable than other bacterial laccases [30,35].

### 2.5. Optimization of Pre-Bleaching Parameters of Laccase

Determining the proper enzyme dose and incubation time is necessary before applying it with other enzymes in the kraft pre-bleaching application. Data revealed that treatment with a laccase dose of 8 IU/mL reduced the kappa number to $17.50 \pm 0.24$ from the initial $19.21 \pm 0.15$ (Figure 7). Similar results were obtained with 10 IU/mL of laccase after 180 min of incubation. In our previous study, xylanase at a dose of 15 IU/mL reduced the kappa number of rice straw pulp by 7.87% [24], whereas laccase at a dose of 8 IU/mL reduced the kappa number of kraft pulp by 8.9%.

### 2.6. Influence of Mixture Enzymes (L + X) on the Biobleaching of Mixed-Wood Kraft Pulp

The use of microbial enzymes significantly reduced the environmental burden caused by the paper and pulp industry and lowered the amount of chemicals used during the bleaching process [25]. In this study, mix-wood kraft pulp was pre-bleached with solo and a mixture of laccase and xylanase (derived from the *Bacillus tequilensis* strain UD-3). The maximal release of reducing sugar, phenolics, and hydrophobic compounds, as well as a decline in kappa numbers, were evaluated from the filtrates collected after treating kraft pulp samples with enzymes in order to evaluate the bio-bleaching conditions. Furthermore, the results reveal that the pretreatment of kraft pulp with an enzyme mixture (L + X)

reduced the quantity of chlorine used by 50% while maintaining the paper's physical and optical properties.

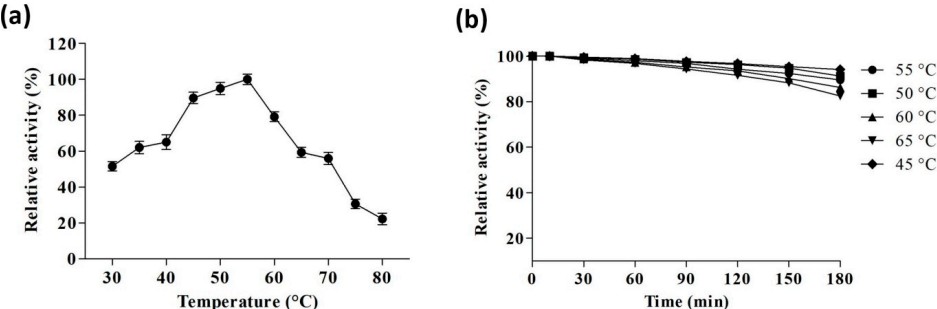

**Figure 6.** Effect of temperature on activity (**a**) and stability (**b**) of the laccase from BK-1 at pH 8.0. (**a**) Laccase activity was measured at different temperatures (30–80 °C). (**b**) Residual activity was measured after pre-incubation of laccase at 45–65 °C for 3 h.

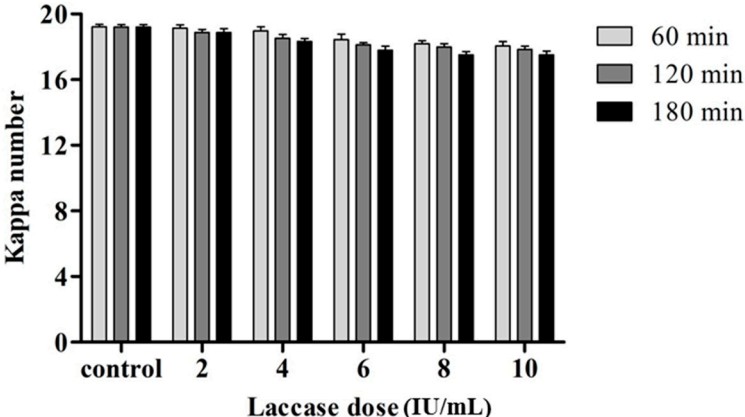

**Figure 7.** Effect of laccase dose (IU/mL) and incubation time (min) on reduction of kraft pulp kappa numbers.

### 2.6.1. Assessment of Filtrate and Pulp Properties after Enzymatic Treatment

After enzymatic treatment, the kappa number significantly decreased (from initial 19.21 to final 15.87), and sugar release was considerably enhanced (from initial 0.079 mg/g to final 2.42 mg/g) (Table 4). Furthermore, enzymatic treatment improved the release of chromophoric compounds such as phenolic, hydrophobic, and lignin compounds (Table 4). Additionally, optical and physical properties such as brightness (43.75%), tear factor (40.55%), burst factor (34.19%), and tensile strength (25.75%) were significantly enhanced after enzymatic treatment (Table 4). This improvement could be attributed to hemicellulolytic component dissolution via xylanase and lignin dissolution via laccase [36,37]. The present study was carried out without using any mediator for laccase. The mediator is a small compound that is constantly oxidized by the laccase and then reduced by the substrate. Many reports suggested that a laccase mediator system is necessary for the laccase use in pulp pre-bleaching [38]. The expense of the mediator system is a significant barrier to their use [38]. However, some reports used laccase without a mediator for pulp biobleaching, including Laccase from *B. tequilensis* SN4 [34] and *B. tequilensis* LXM 55 [6].

**Table 4.** Effect of enzyme mixture (L + X) treatment on mix-wood kraft pulp.

| Parameters | Control | L + X |
|---|---|---|
| Absorbance at wavelengths (nm) | | |
| 237 | 0.058 ± 0.005 | 1.74 ± 0.01 |
| 280 | 0.064 ± 0.008 | 1.98 ± 0.32 |
| 400 | 0.056 ± 0.004 | 1.45 ± 0.28 |
| 465 | 0.031 ± 0.011 | 1.25 ± 0.07 |
| Reducing sugar (mg/g) | 0.079 ± 0.017 | 2.42 ± 0.19 |
| Kappa number | 19.21 ± 0.12 | 15.87 ± 0.08 |
| Brightness (%) | 22.56 ± 0.23 | 32.43 ± 0.15 |
| Tear index (mN m$^2$/g) | 3.23 ± 0.08 | 4.54 ± 0.36 |
| Burst index (kPa m$^2$/g) | 2.31 ± 0.23 | 3.10 ± 0.41 |
| Tensile strength (kN/m) | 15.8 ± 0.25 | 19.87 ± 0.35 |

Note: L + X, laccase–xylanase mixture.

The use of a single enzyme in pulp bleaching results in comparatively less improvement in pulp properties [39–41]; it is highly recommended to use the cocktails of enzymes as a pre-bleaching booster. Because the xylanase mixed with laccase increases the accessibility of residual lignin, facilitates the removal of hexenuronic acids, reduces the use of mediators, and shortens the reaction time by half, the use of an enzyme mixture for bleaching pulp improved its properties rapidly and effectively [1,4,6,42].

### 2.6.2. Determination of Optical and Physical Properties after Enzymatic and Chlorine-Based Treatment

The optical and physical properties of enzymatically (L + X) treated mix-wood kraft pulp improved significantly after chlorine-based bleaching, including brightness, tear factor, burst factor, and tensile strength (Table 5). The brightness of enzyme-treated pulp increased by 10.39% with 100% chlorine added, indicating that the cellulose chain polymer was not adversely affected. The tear factor, burst factor, and tensile strength all increased after bio-bleaching by 39.77%, 22.82%, and 14.28%, respectively. The biobleaching of mix-wood pulp with an enzyme mixture (laccase, xylanase, and mannanase) from *B. tequilensis* LXM 55 increased brightness, whiteness, breaking length, burst factor, and tear factor after chlorine-based bleaching by 11.07%, 16.63%, 53.89%, 45.33%, and 31.75% respectively [15]. Similar findings were reported by Angural et al. [6] in another study, who found that enzymatic treatment with laccase (without a mediator) and xylanase on mixed-wood pulp enhanced brightness (13.21%) and whiteness (3.36%). The application of a mixture of enzymes improved the tensile strength (23.55%), burst index (20.3%), and tear factor (3.17%) of agro pulp [43]. Agrawal et al. [1] found the increment of breaking length, burst factor, and tear factor by 8.5%, 13.4%, and 10.8%, respectively, for plywood veneer soda anthraquinone pulp. Kaur et al. [44] have reported 4.4%, 9.0%, 4.5%, and 11.8% increments in the breaking length, burst factor, tear factor, and viscosity, respectively, for the biobleaching of mixed hardwood kraft pulp.

**Table 5.** Changes in the optical and physical properties of mix-wood kraft pulp after chlorine-based treatment.

| Treatment | Control | Treated L + X | Improvement (%) |
|---|---|---|---|
| Brightness (%) | 77.33 ± 0.52 | 85.31 ± 0.93 *** | 10.39 |
| Tear index (mN m$^2$/g) | 22.78 ± 0.82 | 31.84 ± 0.43 *** | 39.77 |
| Burst index (kPa m$^2$/g) | 12.31 ± 0.44 | 15.12 ± 0.32 *** | 22.82 |
| Tensile strength (kN/m) | 32.77 ± 0.50 | 37.45 ± 0.69 *** | 14.28 |

Note: The Tuky test was conducted between the control and the treatment. Significant codes: $p < 0.001$ ***.

*2.7. Reduction in Chlorine Use*

Due to the high effectiveness of the enzyme mixture, we investigated the possibility of reducing the amount of chlorine used in the pulp that has been treated enzymatically. After being treated with enzymes, the pulp received lower charges of elemental chlorine (100%, 90%, 80%, 70%, 60%, 50%, and 40%). In the current study, nearly 50% less chlorine was used after treating kraft pulp with an enzyme mixture (L + X) to achieve the nearly same 77.35% ISO brightness, 22.43 mN m$^2$/g tear factor, 12.95 kPa m$^2$/g burst factor, and 33.82 kN/m tensile strength as obtained using a chemical process (Table 6). Angural et al. [6] reported a 40% reduction in chlorine consumption while achieving the same paper quality as pulp treated without enzymes but treated with 100% chlorine. Sharma et al. [4] reported a 25% reduction in chlorine used in the enzyme (X + P)-treated pulp while improving the viscosity (4.2%), breaking length (8.5%), burst factor (13.4%), and tear factor (10.8%). Garg et al. [40] reported a 20% reduction in chlorine consumption and a 17.5% reduction in chlorine dioxide consumption after treating wheat straw pulp with xylanase to achieve the same proportion of ISO brightness as the chemical process. Lin et al. [45] reported a 10% reduction in chlorine dioxide consumption for the xylanase-aided bleaching of wheat straw pulp, while xylano-pectinolytic enzymes represented a 25% reduction in chlorine consumption compared to the control with mixed hardwood kraft pulp [39]. The use of mixed enzymes consequently reduces the quantity of chlorine used in the bleaching process, resulting in a greener and cleaner papermaking process. There are numerous advantages to using less hypochlorite in the bleaching process, including less degradation of cellulose fibers, increased paper strength, and decreased dichlorides and ethers produced due to the hypochlorite treatment. The present research result for pre-bleaching kraft pulp with an enzyme mixture is effective, but implementation on an industrial scale requires additional study.

**Table 6.** Reduction in chlorine consumption of mix-wood kraft pulp treated with a mixture of L + X enzymes.

| Chlorine Dose (%) | Untreated Pulp | L + X Treated Pulp | | | | | | |
|---|---|---|---|---|---|---|---|---|
| | 100 | 100 | 90 | 80 | 70 | 60 | 50 | 40 |
| Changed in the optical and physical properties | | | | | | | | |
| Brightness (%) | **77.33** | 85.31 | 84.67 | 83.2 | 80.43 | 78.11 | **77.35** | 74.78 |
| Tear index (mN m$^2$/g) | **22.78** | 31.84 | 30.12 | 27.67 | 25.04 | 23.5 | **22.43** | 20.15 |
| Burst index (kPa m$^2$/g) | **12.31** | 15.12 | 14.48 | 14.11 | 13.85 | 13.46 | **12.95** | 10.54 |
| Tensile strength (kN/m) | **32.77** | 37.45 | 36.2 | 35.85 | 35.1 | 34.33 | **33.82** | 30.63 |

Bold values.

**3. Materials and Methods**

*3.1. Microbial Strain and Enzyme Production Conditions*

A previously isolated bacterial strain, *Bacillus tequilensis* strain UD-3 (Accession no. MT705006), for xylanase production was utilized in the present study. The fermentation conditions utilized for xylanase production were implemented in accordance with the methodology outlined by Patel and Dudhagara [13].

*3.2. Chemicals and Raw Materials*

Guaiacol, 3,5-Dinitrosalicylic acid (DNSA), ethylenedimine tetraacetic acid (EDTA), and hydrogen peroxide were purchased from Hi-media (Thane, India), and all other analytical-grade chemicals were purchased from Loba Chemie (Mumbai, India). Lignocellulosic residues were obtained from the regional farms and markets in Surat, India.

*3.3. Sample Collection and Isolation of a Laccase Producer*

An alkaline hot spring water sample was collected in a sterile plastic bottle from the Bakreshwar hot spring, which is situated in the Birbhum district of West Bengal, India (Latitude: 23°52′52.00″ N, Longitude: 87°22′32.00″ E). Bacterial isolation was carried out

by spreading 0.5 mL of the water sample on the nutrient agar plate, which was incubated at 45 °C until colonies appeared. Morphologically different bacterial isolates were purified via repeating four flame streaking and further screened for laccase activity. The laccase screening process involved applying a pure bacterial culture onto a modified agar medium enriched with 10 mM guaiacol. The culture was then incubated at a temperature of 45 °C for 48 h. The isolates that exhibited the development of a reddish-brown color were identified as laccase positive and then chosen for more investigation.

The detailed composition of the modified agar medium was as follows: guaiacol, 10 mM; glucose, 0.1%; yeast extract, 0.5%; $CuSO_4 \cdot 5H_2O$, 0.001%; $K_2HPO_4$. 0.1%; $MgSO_4 \cdot 7H_2O$, 0.02%; NaCl, 0.2%; $CaCl_2$, 0.01%; agar, 3.0%; and adjusted to pH 7.0 in 100 mL distilled water.

### 3.4. Identification of the Laccase Producer

Laccase-producing bacteria were identified using 16S rRNA gene sequencing. Genomic DNA was isolated according to the method of Wilson [46] and amplified the 16S rRNA gene using the universal primers 27f (5′-AGAGTTTGATCMTGGCTCAG-3′) and 1492r (5′-CGGTTACCTTGTTACGACTT-3′) with maintaining PCR conditions according to Desai and Patel [14]. As part of the Sanger sequencing procedure, a purified PCR amplicon was sequenced at Eurofins Genomic India Pvt Ltd. in Bangalore, India. The obtained sequence was analyzed for sequence matches using NCBI's Nucleotide Basic Local Alignment Search Tool (BLASTn of BLAST+ 2.15.0).

### 3.5. Preparation and Evaluation of the Lignocellulosic Substrate for Laccase Production

Lignocellulosic residues such as rice bran (consisting of approximately 30–35% cellulose, 22% hemicellulose, and 7–10% lignin), wheat bran (consisting of approximately 2–3% cellulose, 11–26% hemicellulose, and 3–5% lignin), oat bran (consisting of approximately 9–12% cellulose, 7–8% hemicellulose, and 2–4% lignin), barley bran (consisting of approximately 26–27% cellulose, 32–33% hemicellulose, and 21% lignin), and sawdust (consisting of approximately 40% cellulose, 20–21% hemicellulose, and 31–32% lignin) were screened as carbon sources for growth and laccase production. Sawdust was oven-dried at 60 °C before being cut into small pieces, crumbled with an electric grinder, and passed through a 0.5 mm filter for uniform particle size [24].

Laccase production via submerged fermentation was carried out in a 250 mL Erlenmeyer flask by adding 1% (*w/v*) to each substrate into a basal medium (devoid of guaiacol). All the flasks were inoculated with 1% (*v/v*) inoculum (absorbance, $A_{600}$ nm; 0.6) and incubated at 45 °C for 96 h. During incubation, 2 mL of culture was harvested at regular time intervals to determine enzyme activity. The collected sample was centrifuged at $10,000 \times g$ for 15 min to remove the debris, and the clear supernatant was used for laccase activity.

### 3.6. Enzyme Assay

The laccase activity was measured by monitoring the oxidation substrate guaiacol. A total of 1 mL of a 10 mM guaiacol solution was buffered with a 3 mL (10 mM) sodium acetate buffer (pH 5.0) and mixed with 1 mL of culture filtrate. The mixture was incubated at 30 °C for 5 min, and the absorbance was measured at $\lambda_{470}$ nm. Enzyme activity was expressed as International Units (IU), where one IU is described as the amount of enzyme required to oxidize 1 µmol of guaiacol per minute under assay conditions. The reaction was performed in duplicate, and average values were used. The laccase activity was calculated using the formula given by Patel et al. [26]:

$$\text{Enzyme activity (IU/mL)} = (A \times V)/(t \times e \times v)$$

where A = absorbance at 470 nm, V = total volume (mL), t = incubation time (min), e = coefficient of guaiacol (26.6 mM × cm/L) at 470 nm, and v = enzyme volume (mL).

### *3.7. Statistical Optimization of Laccase Production*

#### 3.7.1. Plackett–Burman Design

Based on the available literature data, eleven different factors, i.e., A: glucose, B: rice bran, C: yeast extract, D: $K_2HPO_4$, E: $MgSO_4 \cdot 7H_2O$, F: NaCl, G: $CuSO_4 \cdot 5H_2O$, H: $CaCl_2$, J: temperature, K: pH, and L: inoculum sizes were selected and the Placket–Burman (PB) design was employed to select significant factors among them. Each factor was examined at both the high (+) and low levels (−). A design of a total of 12 experiments was generated using the Design Expert® (Stat Ease, Minneapolis, MN, USA) version 12.0 (Table 1), and the result was concluded based on the Pareto chart.

#### 3.7.2. Central Composite Design

Further, the factors that showed a significant positive effect (rice bran, yeast extract, $CuSO_4 \cdot 5H_2O$, and temperature) on the production of laccase were again optimized using a CC design to determine the interaction between significant factors. The minimum and maximum ranges of the selected variables, i.e., rice bran (20 to 35 g/L), yeast extract (15 to 25 g/L), $CuSO_4 \cdot 5H_2O$ (0.015 to 0.025), and temperature (35 to 55 °C) were used in 30 combinations (Table S1). The RSM model was validated further for predicted versus actual responses. The model adequacy and significance were determined using the analysis of variance (ANOVA). The statistical analysis and three-dimensional (3D) graph were generated using the Design Expert version 12 (Stat-Ease).

### *3.8. Enzyme Characterization*

#### 3.8.1. Effect of pH on Laccase Activity and Stability

Enzyme activity was evaluated at pH 4.0–9.0 by incubating the enzyme–substrate mixture with the following buffer systems: a citrate phosphate buffer (pH 4.0 and 5.0), sodium phosphate buffer (pH 6.0 and 7.0), and Tris-HCl buffer (pH 8.0 and 9.0) to determine the optimum pH of the laccase. The relative laccase activity was determined as per the standard assay method. The relative activities were calculated by considering the maximum activity as 100%. The stability of the enzyme at different pH scales was investigated via the pre-incubation of the enzyme solution with different buffers (aforementioned buffers for pH 6.0–8.0) for 3 h at 45 °C temperature, and residual laccase activity was examined in comparison to initial activity as per the standard assay method.

#### 3.8.2. Effect of Temperature on Laccase Activity and Stability

The optimal temperature of the laccase activity was determined by performing enzymatic assays in a sodium phosphate buffer (pH 7.0) at temperatures ranging from 30 to 80 °C for 10 min. The relative activities were calculated by assuming the maximal activity to be 100%. The thermal stability of the laccase was determined by pre-incubating the enzyme at temperatures ranging between 45 and 65 °C along with a sodium phosphate buffer (pH 7.0) for 3 h. The residual laccase activity was compared to the initial activity as per the standard assay method.

### *3.9. Optimization of Laccase Enzyme Pre-Bleaching Conditions on Kraft Pulp*

The laccase was initially investigated for optimizing kraft pulp pre-bleaching conditions. Unbleached pulp samples were thoroughly washed with tap water, oven-dried, and used for further study [13]. The pulp was treated with different doses of laccase (2–10 IU/mL) at different time intervals (60–180 min) at an adjusted pH of 8.0. The entire process was carried out in polyethylene bags at 50 °C. After the process was completed, the lignin content was calculated by reducing the kappa number.

### *3.10. Determination of the Biobleaching Effect of Laccase and Xylanase on Kraft Pulp*

The bio-bleaching experiments were initiated using a 5% mix-wood kraft pulp consistency and treated with a mixture of optimized laccase (8.0 IU/mL) and previously optimized xylanase (15 IU/mL). Incubations were set at pH 7.0 and 50 °C for 3 h in

polyethylene bags for the pre-bleaching treatment. Kraft pulp treated under the same conditions but without enzymatic treatment was taken as the control. After the enzymatic pre-bleaching treatment, the bio-bleached pulp was chemically bleached using hypochlorite, followed by EDTA and hydrogen peroxide treatment. Various concentrations (50 to 100%) of hypochlorite were prepared, and each portion of the pulp sample was treated with a respective hypochlorite concentrate for 1 h at 50 °C. Pulp samples were washed and air-dried at the end of the first chemical treatment process. Then, all samples were treated with a 1% EDTA solution at 90 °C for 1 h, followed by 0.5% hydrogen peroxide in a consecutive step with the same incubation condition used in the previous step. Pulp samples were filtered through muslin cloth, washed with distilled water, and then oven-dried.

### 3.11. Analysis of Filtrate Properties

The chromophoric compounds were measured at 237, 280, 400, and 465 nm using a spectrophotometer [47]. Reducing sugars were estimated using the DNAS method [48]. Kappa numbers were determined by performing the standard TAPPI T 236 cm 85 method [49].

### 3.12. Sheet Preparation and Analysis of Optical and Physical Properties

Hand sheets of control pulp and enzyme-treated pulp were prepared using T205 sp-02 standard TAPPI methods to examine the optical and physical properties of the pulp [45]. Optical properties, i.e., rightness methods, were measured using T 452 OM-87, whereas physical properties like the tensile strength, tear index, and burst index were determined using T 404 cm-92, T 414 OM-98, and T 403 OM-97 methods, respectively. Each experiment was run twice, with each repetition using a sample in triplicate.

## 4. Conclusions

Bakreshwar hot spring's thermophilic isolate was found to be an excellent source of moderately thermostable laccase. Laccase could be produced from a variety of agricultural waste materials. However, rice bran and sawdust were found to be the most suitable. Rice bran has been successfully valorized for the production of laccase. Furthermore, statistical optimization increased laccase production by approximately 2.5 times to develop direct crude laccase application for the kraft pulp process. Laccase was useful for pre-bleaching kraft pulp because of its thermostability and stability in a wide pH range [50–52]. The commercial significance of low-cost laccase with a cheap substrate has enabled paper mills to adopt enzyme-based pulp-bleaching techniques. The current investigation found that the enzyme mixture (L + X) decreased chlorine by 50% in mixed-wood kraft pulp without compromising the paper quality. Both enzymes are stable at a wide range of pH and temperature, making them an excellent choice for the pre-bleaching procedure. Pre-bleaching using an enzyme mixture can efficiently reduce hemicellulolytic components via xylanase action, dissolve lignin via laccase, and remove lignin from kraft pulp while enhancing the necessary physiochemical characteristics for paper manufacture. Furthermore, the cost of effluent treatment will be lowered due to decreased chlorine and water use and, as a result, decreased formation of dichloride and ethers. Laccase offers a valuable and environmentally friendly alternative to traditional chemical bleaching methods in the pulp and paper industry. Its cost-effectiveness, compatibility with existing processes, and ability to improve paper quality while reducing environmental impact make it a promising technology for the future of sustainable paper production. This research provides valuable information for the pulp and paper industry to adopt enzyme-based pre-bleaching processes and reduce hazardous chemicals with less water use.

**Supplementary Materials:** The following supporting information can be downloaded at https://www.mdpi.com/article/10.3390/catal14010001/s1. Table S1: Variables optimized by central composite (CC) design for enhanced production of laccase.

**Author Contributions:** K.P.: conceptualization, methodology, investigation, resources, writing—original draft, visualization; N.V.: data curation; K.S.: data curation; S.M.D.: data curation; A.G.: data curation; P.D.: validation, writing—review and editing; D.J.H.S.: writing—review and editing, project administration, supervision. All authors have read and agreed to the published version of the manuscript.

**Funding:** The authors are thankful to the Gujarat State Biotechnology Mission, Gandhinagar, for funding support under the Research Support Scheme (Project no: 0RPC64). KP was supported by a National Fellowship for Other Backward Classes (NFOBC) (Candidate ID: NFO-2018-19-OBC-GUJ-67289) from the University Grant Commission (UGC), New Delhi, India.

**Data Availability Statement:** The data presented in this study are available on request.

**Acknowledgments:** The authors are grateful to Agarwal Paper Mills for analyzing pulp properties and the support resources provided by the National Pingtung University of Science and Technology.

**Conflicts of Interest:** The authors declare no conflict of interest.

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
