# Peer review of "Synergistic Use of Thermostable Laccase and Xylanase in Optimizing the Pre-Bleaching of Kraft Pulp"

_catalysts, doi:10.3390/catal14010001_

Round 1
Reviewer 1 Report
Comments and Suggestions for Authors
The reviewed manuscript presents an applied study aimed at optimizing an enzymatic process that could be efficient for a pre-bleaching treatment of kraft pulp in order to limit subsequent use of chemical bleaching agents. A large part of the manuscript consists in finding the best conditions for production (nutrient source, culture parameters) and use of laccase (pH, temperature, treatment time) one of the key enzymes, from a Bacillus strain isolated from a hot-spring water, combining statistical methods and experiments. This classical study is rigorously conducted. The last part of the manuscript describes “real” experiments of the pre-bleaching process comparing the effect of each enzyme alone and the mixture of the characterized laccase and xylanase. This xylanase is said to come from the same bacterial strain in the Abstract and from another strain in the M&M section. Table 4, that is supposed to present these results, is incomplete and the conclusion is therefore not obvious. Moreover, the parameters measured are not always easy to interpret for non-specialist reader (absorbances at different wavelengths, kappa number, brightness, tear index, burst index, tensile strength) and should be defined (in some cases, at least). For all the results, the errors made during the experiments or due to the apparatus should be taken into account and integrated in the decimal number of digits presented (for most of them, one decimal digit should be enough!). Finally, this manuscript should be subjected to a serious proofreading in order to enhance its quality.
Introduction
· L76-82: The authors present the enzymatic mixtures previously described in the literature for pulp bleaching but they do not give any results from these studies or overview conclusion in order to explain their objectives and why they choose this mixture.
Results and discussion
· L91: How was the isolate BK1chosen among the 8 found? On what criteria?
· L92: The end of sentence “and which .. medium” is incorrect. Modify.
· Section 2.2. should be read carefully and modified in order to improve the understanding and logic.
· Correct the unit symbol of l to L throughout the text.
· L116: The increase of laccase production and then decrease is observed with all the substrates and can not be described only for rice bran. Do the authors have an explanation of the decrease? Have the authors monitored the biomass weight during incubation?
· Reference 19 is cited before Reference 20.
· L118-119: What is the origin of the laccase described in reference 20?
· L131: Define PB
· Table 1. The factors instead of letters should be clearly indicated in this Table. Be careful: The numbers of lines are superimposed on the number of run.
· Figure 3: (a) The factors are written too small to be readable. (b) The factors are not indicated on the last three bars. Are the two graphs (a and b) really necessary? What information does each one provide?
· L159: Define CC.
· L165: Explain “the process variables within their range”.
· Table 2; What is the meaning of the figures in the table?
· L216: The sources of carbon and nitrogen can certainly affect the growth of Bacillus licheniformis, the enzyme production being linked to the cell number. Have the authors some information about the evolution of OD or biomass weight?
· L225: Same question here: do- low temperatures decrease microbe efficiency or microbe growth?
· L268: “but more thermostable … laccases”. Not understandable.
· Figure 7: What is the unit of “Laccase dose”?
· L298 and Table 4: Give more explanation on the choice of the wavelengths and the link to “phenolic and hydrophobic and lignin compounds”?
· Table 4: The columns with the results of L and X alone are missing. The interest of kappa number measurement should be explained.
· Table 5: Does the control correspond to the chlorine based treatment without any pre-treatment?
Materials and methods
· L375: Why do the authors put an “s” to strains?
· L383: Replace “lingocellulose” with “lignocellulose”
· L414: Do the “all residues (excluding brans)” correspond only to sawdust? If it is the case, modify the sentence.
· L428: Why do the authors give the complete name of guaiacol here as they have cited this chemical several times?
· L438: The meaning of “M” unit is mol/L. Remove /L.
· L477: Remove “under” before “for”.
· L513: Change “4.12” to “3.12”.
In the Conclusion, the environmental impact of this “new” process could be more emphasized (as well as the cost of the water treatment) by the limitation of chlorine use.
Comments on the Quality of English Language
The manuscript should be subjected to a serious proofreading in order to enhance its quality.
Reviewer 2 Report
Comments and Suggestions for Authors
In this study, Patel et al. investigated the synergistic use of laccase and xylanase derived from a hot-spring Bacillus licheniformis strain to optimize a pre-bleaching process on kraft pulp. The authors isolated the laccase-producing bacterium Bacillus licheniformis BK-1 from the Bakreshwar hot spring in India and tested it for laccase production using different lignocellulosic substrates. It was demonstrated that the maximum laccase production was 8.25 IU/ml in the presence of rice bran as a substrate, followed by 5.14 IU/ml using sawdust over a 48-hour period. They also tested xylanase, previously isolated by the same group. The findings revealed that the co-application of these enzymes can reduce chlorine use by 50% in mixed-wood kraft pulp without compromising paper quality. Since both enzymes maintain their activity over a wide range of pH and temperature, they are excellent candidates for the pre-bleaching process. While these enzymes have been widely studied in the literature, the authors focused on their synergistic effect. The study's purpose is important for the field as it contributes to both economic and environmental aspects. The manuscript is well-designed and conducted, and the results were statistically analyzed to demonstrate the difference between applying the enzymes individually and together.
These minor comments need to be addressed before the publication.
- The title is lengthy and confusing. For instance, 'Synergistic Use of Laccase and Xylanase in Optimizing Pre-Bleaching of Kraft Pulp'.
- Figure 4 requires a more detailed discussion.
- Section 2.4 appears short and simplistic as a separate section. It requires enhancement and elaboration.
- Statistical analysis should be conducted for the data presented in Table 5.
- The conclusion section should summarize the findings, emphasizing their significance for the field and outlining future directions. While the results are highly beneficial for the industry, the conclusion should emphasize their importance more explicitly.
Reviewer 3 Report
Comments and Suggestions for Authors
The article proposed by Patel and coworkers present interesting advances in pre-bleaching pulp process thank to the use of laccase and xylanase. Some improvements can be conducted to enhance the overall quality of this work.
The abstract and the introduction are clear and well written allowing the reader to understand well the aim of this work. However, in abstract the eighth sentence could be improved (l30-32) by starting with the content of chlorine. In the first reading, the improvement of the pulp seems to be at 50% content of chlorine. In the same manner, l80-82, why laccase-xylanase appears twice?
In the result and discussion part (l.119) it will be appraciated to indicate the time at which time 4.58IU/mL were recorded. Indicate the full name of the design for screening (PB) and optimization (CC). Table 1 can not be understood by itself. It will be more helpfull for the reader to invert tables in the SI with the matrices in the manuscript (also for optimization). In addition a reference should be added to justify the recommended level of CuSO4 (l.220).
Furthemore, in Figure 3 information about parameters are lacking. With the appropriated table in the manuscript, in figure 3 only letter could be used.
For the optimisation, an optimum should be searched thank to the equation to find maybe even better reaction conditions. An these reactions should be tested to validate the model by comparing the experimental value with the predicted one.
The effect of pH and temperature should be presented before the optimisation to explain why temperature is not higher than 65°C. The number of significant number is over estimated in this section. I don't think that 100 replicats have been done for the measurement of each enzymatic activity.
Figure 7, a statistical analysis will reinforced the choice of 8 IU/mL. Currently, it is not clear how different is 6 to 8 UI/mL for exemple.
Section 2.8 is the introduction of section 2.9 and 2.10, fix this. In addition the last sentence of current section 2.8 is the conclusion of this section and should be placed at is "real" end.
Footnotes of table 4 include L and X whereas only L+X results were given. For discussion, when compare to the litterature, focus on one parameter then the following one to help reader to follow. In addition presenting result of tests performed by others teams while you don't have the value is not recommended. Indeed, what is the improvement of whiteness for your treatment?
Methods part should include the number of replicate (Enzyme assay). As previously mentionned, if table in supporting information are not placed in the manuscript, levels of the factor should be indicated.
Why pH was changed between pre-bleachin with laccase (pJ 8.0) and with cocktail laccase-xylanase (pH 7.0)?
Comments on the Quality of English Language
l60. A l is missing for "pulp"
l.163 matrix singular
l251. "range of pH may be due" instead of "range of the pH is may be due"
l.430 "and the absorbance measured" instaed of "and measured the absorbance"
l464. evaluated at pH 4.0-9.0 should be between pH 4.0-9.0
l.477 under for 10 min
l.479 ranging between 45-65°C
Reviewer 4 Report
Comments and Suggestions for Authors
Review (recommended major revisions)
Article describes the production of decomposing laccase enzyme, its serial use for bio-bleaching and discusses the latter. In my personal opinion, article is interesting, describes the valorisation of the bran for the production of laccases and provides discussion.
***uploaded review file***

Comments on the Quality of English LanguageReview (recommended major revisions)
Article describes the production of decomposing laccase enzyme, its serial use for bio-bleaching and discusses the latter. In my personal opinion, article is interesting, describes the valorisation of the bran for the production of laccases and provides discussion.
***uploaded review file***
Round 2
Reviewer 4 Report
Comments and Suggestions for Authors
Accept.